# PARAMETER-EFFICIENT FINE-TUNING VIA CIRCULAR CONVOLUTION

## ABSTRACT

Low-Rank Adaptation (LoRA) has gained popularity for fine-tuning large foundation models, leveraging low-rank matrices $\mathbf{A}$ and $\mathbf{B}$ to represent weight changes (*i.e.,* $\Delta\mathbf{W} = \mathbf{BA}$). This method reduces trainable parameters and mitigates heavy memory consumption associated with full delta matrices by sequentially multiplying $\mathbf{A}$ and $\mathbf{B}$ with the activation. Despite its success, the intrinsic low-rank characteristic may limit its performance. Although several variants have been proposed to address this issue, they often overlook the crucial computational and memory efficiency brought by LoRA. In this paper, we propose Circular Convolution Adaptation ($\text{C}^3\text{A}$), which not only achieves high-rank adaptation with enhanced performance but also excels in both computational power and memory utilization. Extensive experiments demonstrate that $\text{C}^3\text{A}$ consistently outperforms LoRA and its variants across various fine-tuning tasks.

## 1 INTRODUCTION

In recent years, Large Foundation Models (LFMs) have witnessed a pronounced ascendance in both scholarly and practical realms, attributable to their exceptional efficacy across diverse tasks in natural language processing (NLP) (Brown et al., 2020; Touvron et al., 2023), computer vision (CV) (Radford et al., 2021; Kirillov et al., 2023), and other domains (Li et al., 2024). Distinguished by an extensive parameter count and significant computational requisites, these models have established unprecedented benchmarks in both accuracy and versatility. Nonetheless, their considerable size and intricate structure present formidable obstacles for efficient fine-tuning, especially within resource-constrained environments (Malladi et al., 2023; Zhang et al., 2024b). To mitigate these challenges, parameter-efficient fine-tuning (PEFT) techniques (Mangrulkar et al., 2022), exemplified by Low-Rank Adaptation (LoRA) (Hu et al., 2021), have emerged as highly effective solutions.

LoRA reduces the number of trainable parameters by leveraging low-rank matrices to approximate alterations in weights, thereby facilitating fine-tuning without degrading the model's efficacy. Specifically, LoRA can be articulated mathematically as follows:

$$\mathbf{Wx} = (\mathbf{W}_0 + \Delta\mathbf{W})\mathbf{x} = \mathbf{W}_0\mathbf{x} + \mathbf{B}(\mathbf{Ax}),$$

where $\mathbf{W}, \mathbf{W}_0, \Delta\mathbf{W} \in \mathbb{R}^{d_1 \times d_2}$ are weight matrices, $\mathbf{B} \in \mathbb{R}^{d_1 \times r}, \mathbf{A} \in \mathbb{R}^{r \times d_2}$ are low-rank matrices formulated to construct $\Delta\mathbf{W}$, and $\mathbf{x} \in \mathbb{R}^{d_2}$ are the activations. The number of trainable parameters is $r(d_1 + d_2)$, thereby motivating the selection of $r \ll \min(d_1, d_2)$

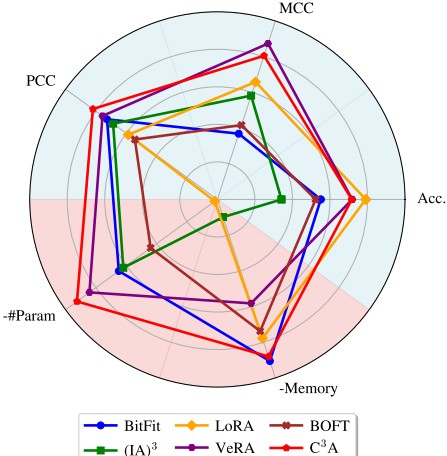

Figure 1: Relative comparison of $\text{C}^3\text{A}$ and baselines on RoBERTa-Base. The Pearson Correlation Coefficient (PCC) is evaluated on STS-B and the Matthew's Correlation Coefficient (MCC) on CoLA. Accuracies across SST-2, MRPC, QNLI, and RTE are averaged and reported as Acc. -#Param shows the reduced number of learnable parameters compared to LoRA, and -Memory indicates the decrease in peak GPU memory usage during training, also compared to LoRA. The metrics in blue pertain to performance-related values, whereas those shadowed in red correspond to values associated with resource consumption. All metrics are the higher the better. See Table 2 for more statistics.

(*e.g.*, $r = 8$ for $d_1 = d_2 = 1024$) to attain elevated parameter efficiency. Nonetheless, as elaborated by Zeng & Lee (2023), the potential of LoRA to encapsulate a target model is inherently constrained by $r$. In an effort to reconcile the dichotomy between performance and efficiency, Kopiczko et al. (2023) introduced Vector Random Matrix Adaptation (VeRA). VeRA attains comparable performance with a markedly reduced count of trainable parameters via fixed random-matrix projections. However, despite its minimal parameter count, VeRA demands considerable computational resources and memory capacity due to the extensive nature of the random matrices employed for projection. As depicted in Figure 1, other representative works share the same resource problem. This precipitates the following open research question within the scope of PEFT:

> *Beyond low parameter counts, how to achieve high-rank adaptation*
> *without incurring significant costs of time and memory?*

To address this question, we introduce Circular Convolution Adaptation ($C^3A$), which incorporates the circular convolution operator (Bamieh, 2018). Circular convolution has garnered significant attention in both signal processing (Li et al., 2020) and cryptography (Dworkin et al., 2001) due to its exceptional efficiency and compactness. This operator can be equivalently expressed as multiplication by a circulant matrix, providing rank flexibility that is independent of the number of trainable parameters. Furthermore, by employing the Fast Fourier Transform (FFT), $C^3A$ achieves superior time and memory efficiency compared to the direct multiplication of the circulant matrix (Bamieh, 2018), which makes it competitive with LoRA in terms of efficiency.

In addition, as explicated by Dosovitskiy et al. (2020), dense linear layers exhibit a deficiency of inductive biases, engendering a complex optimization landscape. Consequently, this hampers the effectiveness of transformers in comparison to Convolutional Neural Networks (CNNs) under conditions of limited data availability. Within the framework of a constrained training dataset for the downstream task, we postulate that a robust inductive bias could potentially augment adaptation performance. The circular pattern in $C^3A$ serves precisely as such an inductive bias.

In summary, circular convolution presents a promising solution for circumventing the rank limitations of LoRA at minimal costs. Our contributions can be summarized as follows:

❶ We introduce $C^3A$, a novel approach for PEFT. This method leverages the circular convolution operation and its equivalent circulant matrix to provide a flexible rank, which is free of linear constraint by the number of trainable parameters, for the delta matrix.

❷ Leveraging the elegant diagonalization of the circulant matrix, we implement both the forward pass and backpropagation using FFT. With the incorporation of FFT, the computation and memory efficiency of $C^3A$ excels. $C^3A$ strikes a unique balance between performance an efficiency.

❸ To offer greater flexibility in controlling the number of trainable parameters, we extend $C^3A$ by incorporating block-circular convolution, which results in block-circulant matrices. This extension allows $C^3A$ to achieve fully customizable parameter counts as well as adaptable rank configurations.

❹ We validate $C^3A$ through comprehensive fine-tuning experiments across diverse tasks including natural language understanding, instruction tuning and image classification. Experiments demonstrate $C^3A$'s outstanding accuracy and memory merits compared to existing state-of-the-art methods.

## 2  RELATED WORK

### 2.1  PARAMETER-EFFICIENT FINE-TUNING

Research on PEFT has generally progressed along three main directions. The first direction involves partially updating the pre-trained neural network (*e.g.*, the layer norm (Basu et al., 2024) or the biases (Zaken et al., 2021)). Traditional methods relied on hand-crafted heuristics (Raghu et al., 2019) to identify which parameters are crucial and should be fine-tuned. More advanced approaches employ optimization techniques (Guo et al., 2020; Xu et al., 2021; Fu et al., 2023). For example, Guo et al. (2020) reformulated such a discrete optimization problem into a continuous one by employing Bernoulli masks and the Gumbel-softmax approximation (Jang et al., 2016).

The second direction emerged to maintain the integrity of the pre-trained model while enabling a high degree of parameter sharing through adapter-based methods (He et al., 2021; Rebuffi et al.,

2017; Rücklé et al., 2020; Liu et al., 2022; Lian et al., 2022). These works focus on integrating additional modules, termed adapters, to fit the downstream task, effectively decoupling the pre-trained model parameters from those specific to the downstream task. Prompt Tuning (Brown et al., 2020; Gao et al., 2020; Chen et al., 2023; Zhang et al., 2024a) and Prefix Tuning (Li & Liang, 2021; Jia et al., 2022) also fall into this category, despite ignoring potential semantic meanings.

The final direction is characterized by delta-weight-based methods, such as Low-Rank Adaptation (LoRA) (Hu et al., 2021) and Orthogonal Fine-tuning (OFT) (Qiu et al., 2023). These methods bridge the gap between the pre-trained model and the downstream task by adaptive delta weights, which are stored separately while used in combination with the pre-trained weights. This unique design enables disentanglement of the pretrained and downstream-specific weights. Namely, it achieves parameter sharing and preserves the ability to integrate without additional inference cost. LoRA models the delta-weights by an additive matrix while OFT does it by a multiplicative one. To further improve either parameter efficiency or performance, many variants has been proposed for both of the methods (Kopiczko et al., 2023; Liu et al., 2024; 2023; Yuan et al., 2024; Hayou et al., 2024b; Gao et al., 2024). However, these methods can hardly achieve high parameter efficiency and performance without incurring heavy computation and memory usage.

## 2.2 Circular Convolution

Circular convolution has been extensively studied in signal processing (Rabiner et al., 1978; McGillem & Cooper, 1984; Li et al., 2020) and cryptography (Dworkin et al., 2001; Gong et al., 2024). Owing to its computational advantages, circular convolution has also been explored in machine learning for generating long embeddings of high-dimensional data (Yu et al., 2014) and compressing heavily parameterized layers (Cheng et al., 2015; Ding et al., 2017). Remarkably, it achieves these efficiencies without significant performance degradation, which makes it a promising technique for fine-tuning applications.

Despite its success in small neural networks such as LeNet (Cheng et al., 2015), circular convolution has not demonstrated lossless performance in modern large foundational models (LFMs) or even in their base architecture, the transformer. This limitation may be attributed to the conflict between its high intrinsic bias (*i.e.,* the circulant pattern) and the vast amount of data required for training LFMs. Conversely, when fine-tuning LFMs, it is often impractical to collect as much data as needed for training from scratch. In such scenarios, the intrinsic bias of circular convolution could potentially serve as a regularization mechanism, thereby benefiting the optimization process of fine-tuning.

## 3 Method

In this section, we present $C^3A$ (see an overview in Figure 2), a novel PEFT approach based on the circular convolution. $C^3A$ follows LoRA's setting of learning an additive linear operation over the original dense linear transformation. However, instead of using low-rank decomposition and the matrix multiplication operator, $C^3A$ resorts to circular convolution as this additive linear operation. Section 3.1 introduces the notations we use. Section 3.2 discusses the circular convolution operator, its equivalent circulant matrix, and its calculation in the frequency domain. Section 3.3 details an efficient method for backpropagation. Section 3.4 describes block-wise convolution for controlling the number of trainable parameters. Finally, Section 3.5 analyzes the computational complexity.

## 3.1 Notations

The adapted weight matrix, the original weight matrix, and the delta matrix are denoted by $\mathbf{W}$, $\mathbf{W}_0$, and $\Delta\mathbf{W}$, respectively ($\mathbf{W}, \mathbf{W}_0, \Delta\mathbf{W} \in \mathbb{R}^{d_1 \times d_2}$). The activation vector of the previous layer is denoted by $\mathbf{x} \in \mathbb{R}^{d_2}$. The post-transformation vector is $\mathbf{z}$, where $\mathbf{z} = \mathbf{W}\mathbf{x} \in \mathbb{R}^{d_1}$, and the incremental part is denoted by $\Delta\mathbf{z}$, where $\Delta\mathbf{z} = \Delta\mathbf{W}\mathbf{x} \in \mathbb{R}^{d_1}$. The matrices $\mathbf{A}$ and $\mathbf{B}$ are low-rank matrices introduced by LoRA to represent $\Delta\mathbf{W}$, with $r$ being their rank. $r_v$ specifies the rank of the random projection matrix used in VeRA. The circular convolution kernel of $C^3A$ is denoted by $\Delta\mathbf{w}$ and the circular convolution operator by $\star$. The loss function is represented by $\mathcal{L}$. The Fast Fourier Transform and its inverse are denoted by FFT and iFFT, respectively. The Hadamard product is denoted by $\circ$.

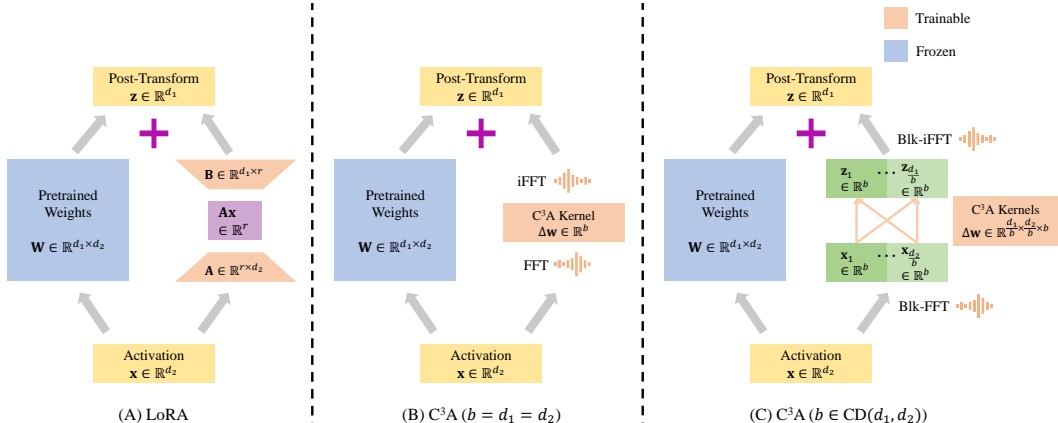

Figure 2: **Overview of LoRA (A) and our $C^3A$ (B,C) method.** In LoRA, only low-rank matrices $\mathbf{A}$ and $\mathbf{B}$ are trained and the delta weight is represented by their product (*i.e.,* $\Delta\mathbf{W} = \mathbf{BA}$). The total trainable parameter number is $r(d_1 + d_2)$, which is assosiated with the rank of the delta weight. In $C^3A$, circular convolution kernels $\Delta\mathbf{w}$ are tuned to adapt to the downstream task and the delta weight is represented by the (block-)circular matrix they construct (*i.e.,* $\Delta\mathbf{W} = \mathcal{C}_{(\text{blk})}(\Delta\mathbf{w})$). The total trainable parameter count is $\frac{d_1 d_2}{b}$, which disentangles with the rank of the delta weight. Here, $b$ is the block size of the block-circular matrix and it should be a common divisor (CD) of $d_1$ and $d_2$.

## 3.2 Circular Convolution

Firstly, for simplicity, we assume $d_1 = d_2 = d$ and $\Delta\mathbf{w} \in \mathbb{R}^d$. The circular convolution operator is defined as $\Delta\mathbf{z} = \Delta\mathbf{w} \star \mathbf{x} = \mathcal{C}(\Delta\mathbf{w})\mathbf{x}$, where $\mathcal{C}(\cdot)$ is a function which takes a vector and outputs the corresponding circulant matrix. Concretely, the first row of $\mathcal{C}(\Delta\mathbf{w})$ is $\Delta\mathbf{w}$ and the following rows are equal to the row above them periodically shifted to the right by one element. In math,

$$
\mathcal{C}(\Delta\mathbf{w}) = \begin{bmatrix}
\Delta w_1 & \Delta w_2 & \cdots & \Delta w_{d-1} & \Delta w_d \\
\Delta w_d & \Delta w_1 & \cdots & \Delta w_{d-2} & \Delta w_{d-1} \\
\cdots & \cdots & \cdots & \cdots & \cdots \\
\Delta w_3 & \Delta w_4 & \cdots & \Delta w_1 & \Delta w_2 \\
\Delta w_2 & \Delta w_3 & \cdots & \Delta w_d & \Delta w_1
\end{bmatrix}.
$$

Theoretically, the rank of $\mathcal{C}(\Delta\mathbf{w})$ is given by $d - \text{Deg}(\gcd(f(x), x^d - 1))$ (Ingleton, 1956), where $\text{Deg}(\cdot)$ denotes the degree of a polynomial, $f(x)$ is the polynomial associated with $\Delta\mathbf{w}$ (*i.e.,* $f(x) = \sum_{i=1}^{d} \Delta w_i x^{i-1}$), and $\gcd(\cdot)$ represents the greatest common divisor. Consequently, the theoretical upper bound on the rank of $\mathcal{C}(\Delta\mathbf{w})$ is $d$. By learning $\Delta\mathbf{w}$ in the $\mathbb{R}^n$ oracle, $C^3A$ automatically achieves dynamic rank selection, which is not linearly constrained by the number of learnable parameters, unlike LoRA.

To achieve high efficiency, enlightened by Ding et al. (2017), we leverage the beautiful circulant structure of $\mathcal{C}(\Delta\mathbf{w})$, which makes it diagonalizable by the Fourier basis ($\mathbf{F}$). In math, it can be described as $\mathcal{C}(\Delta\mathbf{w}) = \mathbf{F}\frac{\Lambda}{d}\mathbf{F}^{-1}$ (Golub & Van Loan, 1996), where $\Lambda$ is its eigenvalues and can be calculated by a Fourier transform of the first row (*i.e.,* $\Lambda = \text{diag}(\mathbf{F}\Delta\mathbf{w})$). Therefore, we can calculate $\Delta\mathbf{w} \star \mathbf{x}$ as

$$
\begin{aligned}
\Delta\mathbf{w} \star \mathbf{x} &= \mathbf{F}\text{diag}(\frac{\mathbf{F}\Delta\mathbf{w}}{d})\mathbf{F}^{-1}\mathbf{x} \\
&= \text{FFT}(\text{FFT}(\Delta\mathbf{w}) \circ \text{iFFT}(\mathbf{x})).
\end{aligned}
\tag{1}
$$

## 3.3 Backpropagation

To effectuate backpropagation with optimal efficiency, it is imperative to obtain the analytical derivatives of the loss function $\mathcal{L}$ with respect to $\Delta\mathbf{w}$ and $\mathbf{x}$. Following the approach outlined in Ding et al. (2017), we aim to explain backpropagation using simpler language. By applying the chain rule, these

derivatives are delineated as follows:

$$\frac{\partial \mathcal{L}}{\partial \mathbf{x}} = \frac{\partial \Delta \mathbf{z}}{\partial \mathbf{x}} \frac{\partial \mathcal{L}}{\partial \Delta \mathbf{z}}, \qquad \frac{\partial \mathcal{L}}{\partial \Delta \mathbf{w}} = \frac{\partial \Delta \mathbf{z}}{\partial \Delta \mathbf{w}} \frac{\partial \mathcal{L}}{\partial \Delta \mathbf{z}}. \tag{2}$$

Given that $\Delta \mathbf{z} = \mathcal{C}(\Delta \mathbf{w})\mathbf{x}$, it logically follows that $\frac{\partial \Delta \mathbf{z}}{\partial \mathbf{x}} = \mathcal{C}(\Delta \mathbf{w})$. Concerning $\frac{\partial \Delta \mathbf{z}}{\partial \Delta \mathbf{w}}$, we observe the commutative property of the circular convolution operation (*i.e.,* $\mathcal{C}(\Delta \mathbf{w})\mathbf{x} = \mathcal{C}(\mathbf{x})\Delta \mathbf{w}$), which implies $\frac{\partial \Delta \mathbf{z}}{\partial \Delta \mathbf{w}} = \mathcal{C}(\mathbf{x})$. Substituting these findings into Equation 2, we derive:

$$\frac{\partial \mathcal{L}}{\partial \mathbf{x}} = \mathcal{C}(\Delta \mathbf{w})\frac{\partial \mathcal{L}}{\partial \Delta \mathbf{z}}, \qquad \frac{\partial \mathcal{L}}{\partial \Delta \mathbf{w}} = \mathcal{C}(\mathbf{x})\frac{\partial \mathcal{L}}{\partial \Delta \mathbf{z}}.$$

These expressions can also be interpreted as circular convolutions:

$$\frac{\partial \mathcal{L}}{\partial \mathbf{x}} = \Delta \mathbf{w} \star \frac{\partial \mathcal{L}}{\partial \Delta \mathbf{z}}, \qquad \frac{\partial \mathcal{L}}{\partial \Delta \mathbf{w}} = \mathbf{x} \star \frac{\partial \mathcal{L}}{\partial \Delta \mathbf{z}}.$$

By meticulously executing this derivative computation in accordance with Equation 1, backpropagation can harness the computational efficacy facilitated by the FFT algorithm.

### 3.4 BLOCK-CIRCULAR CONVOLUTION

Notwithstanding the elegance and efficiency of the circular convolution operator, it is subject to two fundamental limitations stemming from the constraint that the convolution kernel must match the dimensions of the activation vector: ① *It is inapplicable to non-square weight matrices.* ② *The count of learnable parameters remains fixed.* The first restriction hampers its applicability in scenarios such as fine-tuning a LLaMA3-8B model, where the weight matrix dimensions include $4096 \times 1024$. The second constraint diminishes the adaptability of $\text{C}^3\text{A}$, presenting challenges in addressing complex downstream tasks that necessitate a greater number of learnable parameters. To mitigate these limitations, we employ block-circular convolution (Ding et al., 2017). By partitioning the activation vector $\mathbf{x}$ and the post-transformation vector $\Delta \mathbf{z}$ into blocks of identical size, unique convolution kernels can be allocated to each pair of these blocks. Specifically,

$$\mathbf{x} = \begin{bmatrix} \mathbf{x}_1 & \mathbf{x}_2 & \cdots & \mathbf{x}_{\frac{d_2}{b}} \end{bmatrix}, \qquad \Delta \mathbf{z} = \begin{bmatrix} \Delta \mathbf{z}_1 & \Delta \mathbf{z}_2 & \cdots & \Delta \mathbf{z}_{\frac{d_1}{b}} \end{bmatrix},$$

where $b$ is the block size and $b$ need to be a common divisor of $d_1$ and $d_2$. We will need $\frac{d_1 d_2}{b^2}$ convolution kernels to densely connect these blocks, which can be expressed in math as

$$\Delta \mathbf{z}_i = \sum_{j=1}^{\frac{d_2}{b}} \Delta \mathbf{w}_{ij} \star \mathbf{x}_j, i \in \{1, 2, \cdots, \frac{d_1}{b}\}.$$

This calculation can be represented by a block-circular matrix:

$$\Delta \mathbf{z} = \mathcal{C}_{\text{blk}}(\Delta \mathbf{w})\mathbf{x}, \qquad \mathcal{C}_{\text{blk}}(\Delta \mathbf{w}) = \begin{bmatrix} \mathcal{C}(\Delta \mathbf{w}_{11}) & \mathcal{C}(\Delta \mathbf{w}_{12}) & \cdots & \mathcal{C}(\Delta \mathbf{w}_{1\frac{d_2}{b}}) \\ \mathcal{C}(\Delta \mathbf{w}_{21}) & \mathcal{C}(\Delta \mathbf{w}_{22}) & \cdots & \mathcal{C}(\Delta \mathbf{w}_{2\frac{d_2}{b}}) \\ \cdots & \cdots & \cdots & \cdots \\ \mathcal{C}(\Delta \mathbf{w}_{\frac{d_1}{b}1}) & \mathcal{C}(\Delta \mathbf{w}_{\frac{d_1}{b}2}) & \cdots & \mathcal{C}(\Delta \mathbf{w}_{\frac{d_1}{b}\frac{d_2}{b}}) \end{bmatrix}. \tag{3}$$

We refer our readers to Algorithm A1 in Appendix C for a Pytorch implementation. In this context, $\Delta \mathbf{w}_{ij} \in \mathbb{R}^b$, and it follows that $\frac{d_1 d_2}{b^2} b = \frac{d_1 d_2}{b}$ represents the number of learnable parameters. Notably, the parameter $b$ serves as a hyperparameter modulating the quantity of learnable parameters, analogous to the role of $r$ in LoRA. It is imperative to distinguish, however, that whereas $r$ simultaneously governs the rank of the delta matrix and the number of learnable parameters, $b$ exclusively influences the latter. This disentanglement of matrix rank and parameter count facilitates greater adaptability and potentially yields superior outcomes.

### 3.5 COMPLEXITY ANALYSIS

We compare the time complexity and space complexity of LoRA, VeRA and $\text{C}^3\text{A}$ in Table 1. Detailed analysis follows in this section.

### 3.5.1 TIME COMPLEXITY

LoRA integrates low-rank matrices $\mathbf{A}$ and $\mathbf{B}$, which are successively multiplied with the activation vector, resulting in a computational complexity of $\mathcal{O}(r(d_1 + d_2))$. Generally, $r \ll \min(d_1, d_2)$. In contrast, VeRA, despite its high-rank structure and relatively few trainable parameters, suffers from a prohibitive computational complexity of $\mathcal{O}(r_v(d_1 + d_2))$, where $r_v$ can exceed $\max(d_1, d_2)$. Consequently, striking an optimal balance between high rank and computational efficiency remains an elusive task.

On GPUs, the cuFFT backend automatically parallelizes FFT operations along the axes not being transformed, with the degree of parallelism $p$ determined by the available resources. Thanks to the $\mathcal{O}(n \log n)$ complexity of the FFT algorithm used in Equation 1, C³A achieves a time complexity of $\mathcal{O}(\frac{(d_1 + d_2)}{p} \log b + \frac{d_1 d_2}{b})$. The first term is the time complexity for FFT and the second term is for aggregation. In practical scenarios, $b$ is chosen as the greatest common divisor of $d_1$ and $d_2$ to achieve a high compression ratio. Given that, C³A is comparable to LoRA in time complexity.

### 3.5.2 SPACE COMPLEXITY

We analyze the space complexity of LoRA, VeRA, and C³A during training. The differences among these methods primarily arise from the trainable parameters and the auxiliary tensors required for the forward pass and backpropagation. LoRA does not rely on auxiliary tensors,

Table 1: Time and space complexity comparison of LoRA, VeRA and C³A. We split the space complexity into Parameter number and Other auxiliary tensors to help better understand the differences. We highlight that in practice, to achieve similar performance, $\frac{\max(d_1, d_2)}{b} \leq r \ll r_v$.

| Method | Time | # Param | # Other | # Total |
|---|---|---|---|---|
| LoRA | $\mathcal{O}(r(d_1 + d_2))$ | $r(d_1 + d_2)$ | $0$ | $r(d_1 + d_2)$ |
| VeRA | $\mathcal{O}(r_v(d_1 + d_2))$ | $r_v + d_1$ | $r_v(d_1 + d_2)$ | $r_v(d_1 + d_2) + r_v + d_1$ |
| C³A | $\mathcal{O}(\frac{d_1 + d_2}{p} \log b + \frac{d_1 d_2}{b})$ | $\frac{d_1 d_2}{b}$ | $pb$ | $\frac{d_1 d_2}{b} + pb$ |

while VeRA necessitates 2 random projection matrices, with a total size of $r_v(d_1 + d_2)$. Since $r_v$ is by no means negligible, the memory usage of VeRA is significantly larger than that of LoRA.

In terms of C³A, the only additional auxiliary tensor would be of size $pb \leq \min(d_1, d_2)$, which is reserved by the FFT algorithm. By selecting an appropriate $b$, which is often close to the greatest common divisor of $d_1$ and $d_2$, the space complexity of C³A is optimized. Furthermore, because $p$ scales with the available resources, the algorithm inherently manages dynamic memory consumption without additional effort.

## 4 EXPERIMENT

We first experiment on a synthetic dataset to show C³A's superior expressiveness over LoRA. Next, we evaluate C³A in both NLP and CV. For NLP, we show C³A's effectiveness using the GLUE benchmark with RoBERTa-Base and RoBERTa-Large, and fine-tune the LLaMA family models. For CV, we test classification tasks using Vision Transformers (ViTs) on various datasets. Finally, we perform ablation studies on C³A kernel initialization.

### 4.1 SYNTHETIC DATA

**Settings.** We distribute 8 points evenly on a 2D plane as cluster centers and randomly sample 30 points from the 8 corresponding multivariate Gaussian distributions. A 3-layer MLP is then used to classify these point clusters. To compare the expressiveness of 2 types of layers, we replace the middle layer with either a low-rank layer or a circulant layer, ensuring that both layers have the same number of trainable parameters for a fair comparison.

**Results.** The results are presented in Figure 3. We observe that $\text{LoRA}_{r=1}$ struggles with this simple classification task. In contrast, $\text{C}^3\text{A}_{b=128/2}$, despite using the same number of parameters, achieves a perfect classification, comparable to a standard linear layer. This demonstrates the high expressiveness of C³A given the same parameter budget.

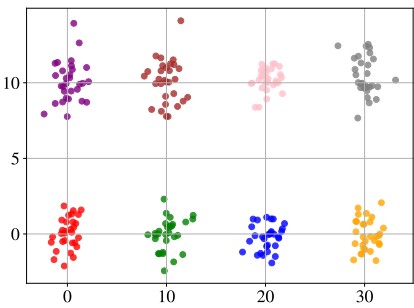 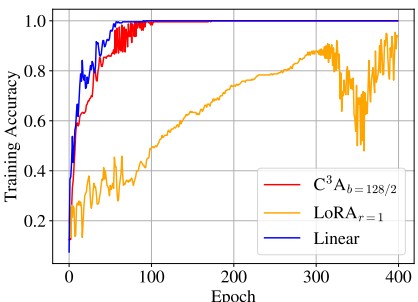

Figure 3: Expressiveness test on synthetic data. The left figure shows the synthetic data used for the experiment, while the right figure illustrates the training accuracy curves of a 3-layer MLP, incorporating $C^3A$, LoRA, and standard linear layers, respectively.

## 4.2 NATURAL LANGUAGE UNDERSTANDING

**Baselines.** We compare our $C^3A$ with several representative PEFT methods, including BitFit (Zaken et al., 2021), $(IA)^3$ (Liu et al., 2022), LoRA (Hu et al., 2021), VeRA (Kopiczko et al., 2023), and BOFT (Liu et al., 2023). BitFit selectively fine-tunes existing parameters, specifically the biases. $(IA)^3$ is the state-of-the-art method that adds additional adapters. LoRA is a widely known PEFT method that employs low-rank decomposition to compress additive delta matrices. VeRA is a recent approach that focuses on further reducing trainable parameters of LoRA while preserving a high rank. BOFT is another innovative method in PEFT research, compressing multiplicative delta matrices using orthogonal decomposition and butterfly factorization.

Table 2: Performance of different PEFT methods on the GLUE benchmark. We fine-tune pre-trained RoBERTa-Base and -Large models on 6 datasets. We report the Matthew's Correlation Coefficient (MCC) for CoLA, Pearson Correlation Coefficient (PCC) for STS-B, and accuracy (Acc.) for all the remaining tasks. For each metric, a higher score indicates better performance. "Avg." denotes the average score of each method across all datasets. The best results for each dataset are highlighted in **bold**. # Trainable parameters does not include the classification head since each method uses a head of the same size. Memory Cost is measured on fixed length (*i.e.,* 256) data with a batchsize of 64.

| | Method | # Trainable Parameters | Memory Cost (GB) | SST-2 | MRPC | CoLA | QNLI | RTE | STS-B | Avg. |
|---|---|---|---|---|---|---|---|---|---|---|
| BASE | Full | 124M | 17.19 | $94.01_{\pm0.39}$ | $\mathbf{87.10}_{\pm0.79}$ | $62.00_{\pm1.16}$ | $92.40_{\pm0.28}$ | $\mathbf{77.33}_{\pm2.68}$ | $\mathbf{90.70}_{\pm0.14}$ | **83.92** |
| | BitFit | 0.102M | 12.60 | $93.30_{\pm0.30}$ | $85.80_{\pm0.21}$ | $59.21_{\pm1.74}$ | $91.96_{\pm0.18}$ | $73.07_{\pm1.34}$ | $90.18_{\pm0.17}$ | 82.25 |
| | $(IA)^3$ | 0.111M | 19.86 | $92.98_{\pm0.34}$ | $85.86_{\pm0.59}$ | $60.49_{\pm1.09}$ | $91.56_{\pm0.17}$ | $69.10_{\pm1.18}$ | $90.06_{\pm0.21}$ | 81.67 |
| | $LoRA_{r=8}$ | 0.295M | 13.75 | $\mathbf{94.50}_{\pm0.41}$ | $85.68_{\pm0.74}$ | $60.95_{\pm1.57}$ | $\mathbf{92.54}_{\pm0.20}$ | $76.68_{\pm1.42}$ | $89.76_{\pm0.39}$ | 83.35 |
| | $VeRA_{r=1024}$ | 0.043M | 15.51 | $93.97_{\pm0.17}$ | $86.23_{\pm0.41}$ | $62.24_{\pm1.91}$ | $91.85_{\pm0.17}$ | $75.74_{\pm1.56}$ | $90.27_{\pm0.25}$ | 83.38 |
| | $BOFT^{m=2}_{b=8}$ | 0.166M | 14.11 | $93.23_{\pm0.50}$ | $84.37_{\pm0.54}$ | $59.50_{\pm1.25}$ | $91.69_{\pm0.12}$ | $74.22_{\pm0.84}$ | $89.63_{\pm0.37}$ | 82.11 |
| | $C^3A_{b=768/1}$ | 0.018M | 12.83 | $93.42_{\pm0.26}$ | $86.33_{\pm0.32}$ | $61.83_{\pm0.96}$ | $91.83_{\pm0.04}$ | $76.17_{\pm1.39}$ | $90.46_{\pm0.29}$ | 83.34 |
| | $C^3A_{b=768/6}$ | 0.111M | 12.72 | $94.20_{\pm0.16}$ | $86.67_{\pm0.54}$ | $\mathbf{62.48}_{\pm1.20}$ | $92.32_{\pm0.25}$ | $77.18_{\pm1.41}$ | $90.16_{\pm0.42}$ | 83.84 |
| LARGE | Full | 354M | 43.40 | $95.75_{\pm0.45}$ | $\mathbf{88.35}_{\pm0.64}$ | $64.87_{\pm1.25}$ | $92.40_{\pm0.28}$ | $84.48_{\pm1.14}$ | $91.65_{\pm0.14}$ | 86.25 |
| | BitFit | 0.271M | 30.65 | $95.09_{\pm0.27}$ | $88.10_{\pm0.76}$ | $65.40_{\pm0.76}$ | $94.06_{\pm0.14}$ | $82.60_{\pm1.15}$ | $91.73_{\pm0.20}$ | 86.16 |
| | $(IA)^3$ | 0.295M | 48.81 | $95.32_{\pm0.20}$ | $87.06_{\pm0.57}$ | $66.52_{\pm1.10}$ | $94.18_{\pm0.15}$ | $84.33_{\pm2.38}$ | $91.58_{\pm0.39}$ | 86.50 |
| | $LoRA_{r=8}$ | 0.786M | 34.12 | $95.53_{\pm0.35}$ | $86.12_{\pm0.86}$ | $65.16_{\pm0.76}$ | $93.73_{\pm0.30}$ | $83.75_{\pm0.51}$ | $91.46_{\pm0.21}$ | 85.96 |
| | $VeRA_{r=256}$ | 0.061M | 34.16 | $\mathbf{95.83}_{\pm0.43}$ | $87.72_{\pm0.55}$ | $63.66_{\pm1.45}$ | $94.11_{\pm0.20}$ | $83.03_{\pm1.65}$ | $91.12_{\pm0.37}$ | 85.91 |
| | $BOFT^{m=2}_{b=8}$ | 0.442M | 34.98 | $95.76_{\pm0.41}$ | $88.28_{\pm0.33}$ | $64.72_{\pm2.37}$ | $93.89_{\pm0.14}$ | $82.82_{\pm1.40}$ | $91.03_{\pm0.32}$ | 86.08 |
| | $C^3A_{b=1024/1}$ | 0.049M | 31.83 | $95.78_{\pm0.05}$ | $88.02_{\pm0.62}$ | $66.59_{\pm1.20}$ | $94.22_{\pm0.25}$ | $82.89_{\pm0.67}$ | $\mathbf{91.86}_{\pm0.14}$ | 86.56 |
| | $C^3A_{b=1024/8}$ | 0.393M | 31.79 | $95.78_{\pm0.15}$ | $88.09_{\pm0.47}$ | $\mathbf{67.18}_{\pm1.92}$ | $\mathbf{94.26}_{\pm0.19}$ | $\mathbf{84.62}_{\pm1.36}$ | $91.81_{\pm0.36}$ | **86.96** |

**Settings.** We evaluate our proposed $C^3A$ on the General Language Understanding Evaluation (GLUE) benchmark (Wang et al., 2018), which encompasses a wide range of natural language understanding (NLU) tasks, including single-sentence classification, similarity and paraphrase, and natural language inference. More dataset specifications can be found in Table A1 in Appendix A. To enhance practicality, we split these datasets following the train-validation-test approach. The best-performing model is selected based on validation set performance across the fine-tuning epochs, and the reported performance corresponds to its performance on the test set. For this evaluation, we fine-tune the pre-trained RoBERTa-Base and RoBERTa-Large models (Liu et al., 2019). For the unique

hyperparameters of different baselines, we adopt the values suggested in the original papers (*e.g.,* VeRA's $r$ and BOFT's $b$ and $m$). The number of trainable parameters excludes the classification head, as each method uses one of the same size. The shared hyperparameters (*i.e.,* the learning rate for classification head and for other trainable parameters, separately) are found by hyperparameter search. For the memory cost, to ensure fairness and consistency, we fix the length of input data to 256 tokens and use a batchsize of 64.

**Results.** The results are presented in Table 2. Overall, $C^3A_{b=768/1}$ and $C^3A_{b=1024/1}$ achieve superior or comparable performance to baseline methods, despite using an exceptionally small number of trainable parameters. As the number of trainable parameters increases, models like $C^3A_{b=768/6}$ and $C^3A_{b=1024/8}$ significantly outperform the baselines. Moreover, compared to $(IA)^3$, LoRA, VeRA, and BOFT, $C^3A$ distinguishes itself with remarkable memory efficiency. The only method demonstrating better memory efficiency is BitFit, which serves as an upper bound since it introduces no new parameters. Additionally, most of the circulant delta matrices identified by $C^3A$ are of full rank, indicating maximal capacity (Zeng & Lee, 2023) and providing a theoretical basis for the outstanding performance.

## 4.3 INSTRUCTION TUNING

**Settings.** For a comprehensive comparison, we further conduct instruction tuning on LLaMA families, the prevalent large language models. Specifically, we evaluate $C^3A$ against LoRA and DoRA (Liu et al., 2024), a variant of LoRA sensitive to learning direction. Specifically, we fine-tune LLaMA2-7B/13B (Touvron et al., 2023) and LLaMA3-8B (Dubey et al., 2024) among 7 datasets covering 3 prevalent tasks: ① Arithmetic reasoning on GSM8k (Cobbe et al., 2021) and MATH (Hendrycks et al., 2020); ② Functional representation generation on ViGGO (Juraska et al., 2019), and SQL (Zhong et al., 2017); and ③ Commonsense reasoning on BoolQ (Clark et al., 2019), PIQA (Bisk et al., 2020) and SIQA (Bisk et al., 2020). For the SQL dataset, we preprocess it by selecting 25% of the data and apply a 4:1 train-test split, resulting in a training set of 16K samples. To ensure a fair comparison, we maintain LoRA parameters with $r = 32$, $\alpha = 32$, and a dropout rate of 0.05, while exploring various learning rates as suggested by (Hu et al., 2021). Please refer to Table A4 in Appendix B for more details.

**Results.** In Table 3, our principal experimental observations are summarized. The $C^3A$ framework consistently surpasses LoRA within the LLaMA series, with particular efficacy demonstrated in the most recent model, LLaMA3-8B. Noteworthy is the significant enhancement in the efficacy of LLaMA3-8B as a foundational model following the implementation of more sophisticated post-training techniques. This underscores the criticality of optimizing the fine-tuning protocols for this advanced model. It is also remarkable that $C^3A$ achieves such results while employing less than half the parameter count of LoRA. Taken together, the findings robustly underscore the superior efficacy of the $C^3A$ methodology. We refer readers to Appendix D for examples of models after different tuning methods.

Table 3: Comparison of $C^3A$ and LoRA on fine-tuning LLaMA2 and LLaMA3 models in terms of accuracy and trainable parameters. The best results for each dataset are highlighted in **bold**. "Avg." denotes the average accuracy of each method across all datasets.

| Model | Method | # Trainable Parameters | GSM8k | MATH | ViGGO | SQL | BoolQ | PIQA | SIQA | Avg. |
|---|---|---|---|---|---|---|---|---|---|---|
| LLaMA2-7B | $LoRA_{r=32}$ | 16.8M | 39.57 | 5.65 | 96.48 | **79.66** | **75.60** | 85.26 | **82.09** | 66.33 |
| | $DoRA_{r=32}$ | 17.0M | 39.05 | **6.00** | 96.85 | 79.66 | 75.41 | 85.64 | 81.93 | 66.36 |
| | $C^3A_{b=4096/32}$ | 8.4M | **40.18** | **6.00** | **97.05** | 79.28 | 75.02 | **85.53** | 81.62 | **66.38** |
| LLaMA2-13B | $LoRA_{r=32}$ | 26.2M | 49.02 | 8.55 | 97.10 | 79.97 | **77.09** | 87.36 | **83.21** | 68.90 |
| | $DoRA_{r=32}$ | 26.6M | **50.02** | **9.00** | 97.32 | 79.66 | 77.16 | 87.70 | 82.60 | 69.07 |
| | $C^3A_{b=5120/32}$ | 13.1M | 49.66 | 8.85 | **97.34** | **80.12** | 76.91 | **87.98** | 83.05 | **69.13** |
| LLaMA3-8B | $LoRA_{r=32}$ | 13.6M | 62.80 | 21.05 | 96.50 | 80.61 | **77.37** | 89.72 | 82.19 | 72.89 |
| | $DoRA_{r=32}$ | 13.8M | 62.95 | **22.15** | 96.54 | **81.22** | 77.09 | 90.21 | 82.44 | 73.24 |
| | $C^3A_{b=4096/32}$ | 5.2M | **64.22** | 21.60 | **96.58** | 80.73 | 77.04 | **90.33** | 82.60 | **73.30** |

## 4.4 IMAGE CLASSIFICATION

**Settings.** In this study, we concentrate on the task of image classification leveraging Vision Transformer (ViT) models. Specifically, we employ both the Base and Large variants of this prominent foundational computer vision model, as delineated by (Dosovitskiy et al., 2020). These ViT models undergo pre-training on the expansive ImageNet-21K dataset (Ridnik et al., 2021). During the fine-tuning phase, we use an eclectic array of datasets encompassing Pets (Parkhi et al., 2012), Cars (Krause et al., 2013), DTD (Cimpoi et al., 2014), EuroSAT (Helber et al., 2019), FGVC (Maji et al., 2013), and RESISC (Cheng et al., 2017). Comprehensive statistics for these datasets are provided in Table A2 in Appendix A.

Table 4: Fine-tuning results with ViT-Base and ViT-Large models on various image classification datasets. The models are fine-tuned for 10 epochs, and the best-performing model, based on validation set accuracy, is selected. The reported accuracy corresponds to the performance on the test set. The best results between LoRA and $C^3A$ for each dataset are highlighted in **bold**. "Avg." denotes the average accuracy of each method across all datasets.

| | Method | # Trainable Parameters | Pets | Cars | DTD | EuroSAT | FGVC | RESISC | Avg. |
|---|---|---|---|---|---|---|---|---|---|
| BASE | Head | - | $90.28_{\pm0.43}$ | $25.76_{\pm0.28}$ | $69.77_{\pm0.67}$ | $88.72_{\pm0.13}$ | $17.44_{\pm0.43}$ | $74.22_{\pm0.10}$ | 61.03 |
| | Full | 85.8M | $92.82_{\pm0.54}$ | $85.10_{\pm0.21}$ | $80.11_{\pm0.56}$ | $99.11_{\pm0.07}$ | $61.60_{\pm1.00}$ | $96.00_{\pm0.23}$ | 85.79 |
| | LoRA$_{r=16}$ | 0.59M | $93.76_{\pm0.44}$ | $78.04_{\pm0.33}$ | $78.56_{\pm0.62}$ | $98.84_{\pm0.08}$ | $\mathbf{56.64_{\pm0.55}}$ | $\mathbf{94.66_{\pm0.17}}$ | 83.42 |
| | $C^3A_{b=768/12}$ | 0.22M | $\mathbf{93.88_{\pm0.22}}$ | $\mathbf{79.05_{\pm0.35}}$ | $\mathbf{80.57_{\pm0.53}}$ | $\mathbf{98.88_{\pm0.07}}$ | $54.31_{\pm0.79}$ | $94.54_{\pm0.23}$ | $\mathbf{83.54}$ |
| LARGE | Head | - | $91.11_{\pm0.30}$ | $37.91_{\pm0.27}$ | $73.33_{\pm0.26}$ | $92.64_{\pm0.08}$ | $24.62_{\pm0.24}$ | $82.02_{\pm0.11}$ | 66.94 |
| | Full | 303M | $94.30_{\pm0.31}$ | $88.15_{\pm0.50}$ | $80.18_{\pm0.66}$ | $99.06_{\pm0.10}$ | $67.38_{\pm1.06}$ | $96.08_{\pm0.20}$ | 87.53 |
| | LoRA$_{r=16}$ | 1.57M | $\mathbf{94.62_{\pm0.47}}$ | $\mathbf{86.11_{\pm0.42}}$ | $80.09_{\pm0.42}$ | $\mathbf{98.99_{\pm0.03}}$ | $63.64_{\pm0.83}$ | $95.52_{\pm0.21}$ | 86.56 |
| | $C^3A_{b=1024/16}$ | 0.79M | $94.48_{\pm0.30}$ | $84.94_{\pm0.39}$ | $\mathbf{82.62_{\pm0.52}}$ | $98.75_{\pm0.19}$ | $\mathbf{63.80_{\pm0.37}}$ | $\mathbf{95.94_{\pm0.16}}$ | $\mathbf{86.69}$ |

**Results.** Table 4 delineates a comprehensive summary of the outcomes derived from six distinct image classification datasets employing the ViT Base and Large models. The LoRA and $C^3A$ techniques exhibit significant enhancements in performance relative to Head Tuning, thereby underscoring their efficacy within the realm of image classification. Remarkably, our methodology demonstrates a performance on par with LoRA while necessitating only half of the parameter count.

## 4.5 INITIALIZATION STUDY

LoRA is known to be sensitive to initialization, primarily due to the distinct roles of its matrices $\mathbf{A}$ and $\mathbf{B}$ (Hayou et al., 2024a). In contrast, $C^3A$ possesses a simpler structure based on circulant matrices, which may reduce sensitivity to initialization. To investigate this, we focused on the initialization strategies for the convolution kernels that define the circulant matrices in $C^3A$. We conducted experiments comparing four initialization methods: zero initialization, Gaussian initialization, Kaiming uniform, and Xavier uniform. We observe that the variations across different initializa-

Table 5: Performance of $C^3A$ with Different Initialization Strategies. The tasks on CoLA and STS-B were performed using the RoBERTa-Base model, while those on Cars and DTD utilized the ViT-Base model. All other settings are consistent with those in Table 2 and Table 4.

| Task | Zero | Gaussian | Kaiming | Xavier | Range |
|---|---|---|---|---|---|
| CoLA | $60.95_{\pm0.88}$ | $61.07_{\pm1.09}$ | $60.82_{\pm1.48}$ | $62.48_{\pm0.74}$ | 1.66 |
| STS-B | $90.23_{\pm0.23}$ | $90.13_{\pm0.16}$ | $90.19_{\pm0.34}$ | $90.31_{\pm0.31}$ | 0.18 |
| Cars | $78.70_{\pm0.60}$ | $78.64_{\pm0.67}$ | $79.18_{\pm0.37}$ | $78.96_{\pm0.25}$ | 0.54 |
| DTD | $80.82_{\pm0.86}$ | $79.58_{\pm0.41}$ | $79.76_{\pm1.14}$ | $79.95_{\pm0.72}$ | 1.24 |

tion points are mostly within the intrinsic standard deviations, highlighting the robustness of $C^3A$ to initialization strategies. Our findings indicate that $C^3A$ maintains robust performance across these different initialization strategies, highlighting its resilience to initialization points.

## 5 CONCLUSION

In this manuscript, we present $C^3A$, a novel method for Parameter-Efficient Fine-Tuning (PEFT). In contrast to LoRA, which employs low-rank decomposition, $C^3A$ leverages circular convolution and its equavelent circulant matrix to represent the delta weight matrix. This methodology aims

to independently control the rank of the delta weight matrix and the number of trainable parameters, facilitating high-rank adaptation while preserving a constrained parameter size. Using the Fast Fourier Transform (FFT) during both the forward and backward propagation phases, $C^3A$ attains notable computational and memory efficiency. In short, $C^3A$ emerges as a persuasive alternative to LoRA for model fine-tuning.

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

# APPENDIX

## A DATASET DETAILS

Table A1: Task descriptions and dataset statistics of the GLUE benchmark (Wang et al., 2018).

| Corpus | Task | # Train | # Val | # Test | # Labels | Metrics | Domain |
|--------|------|---------|-------|--------|----------|---------|--------|
| | | | Single-Sentence Tasks | | | | |
| CoLA | Acceptability | 8.55k | 1.04k | 1.06k | 2 | Matthews Corr. | misc. |
| SST-2 | Sentiment | 67.3k | 872 | 1.82k | 2 | Accuracy | Movie reviews |
| | | | Similarity and Paraphrase Tasks | | | | |
| MRPC | Paraphrase | 3.67 | 408 | 1.73k | 2 | Accuracy/F1 | News |
| STS-B | Sentence similarity | 5.75k | 1.5k | 1.38k | 1 | Pearson/Spearman Corr. | misc. |
| QQP | Paraphrase | 364k | 40.4k | 391k | 2 | Accuracy/F1 | Social QA |
| | | | Inference Tasks | | | | |
| MNLI | NLI | 393k | 19.65k | 19.65k | 3 | Accuracy | misc. |
| QNLI | QA/NLI | 105k | 5.46k | 5.46k | 2 | Accuracy | Wikipedia |
| RTE | NLI | 2.49k | 277 | 3k | 2 | Accuracy | News & Wikipedia |

Table A2: Details about the vision datasets.

| Dataset | #Train | #Validation | #Test | #Class | Rescaled resolution |
|---------|--------|-------------|-------|--------|---------------------|
| Pets (Parkhi et al., 2012) | 3,312 | 368 | 3,669 | 37 | |
| Cars (Krause et al., 2013) | 7,329 | 815 | 8,041 | 196 | |
| DTD (Cimpoi et al., 2014) | 4,060 | 452 | 1,128 | 47 | $224 \times 224$ |
| EuroSAT (Helber et al., 2019) | 16,200 | 5,400 | 5,400 | 10 | |
| FGVC (Maji et al., 2013) | 3,000 | 334 | 3,333 | 100 | |
| RESISC (Cheng et al., 2017) | 18,900 | 6,300 | 6,300 | 45 | |

## B HYPERPARAMETERS

Table A3: Hyperparameter setup of $C^3A$ for the GLUE benchmark.

| Model | Hyperparameter | SST-2 | MRPC | CoLA | QNLI | RTE | STS-B |
|-------|----------------|-------|------|------|------|-----|-------|
| Both | Optimizer | | | AdamW | | | |
| | LR Schedule | | | Linear | | | |
| | Warmup Ratio | | | 0.06 | | | |
| | $C^3A$ Initialization | | | Xavier Uniform | | | |
| | Max Seq. Len | | | 512 | | | |
| Base | Epochs | 40 | 80 | 80 | 40 | 80 | 80 |
| | Batch Size | 128 | 128 | 128 | 64 | 64 | 128 |
| | Learning Rate ($C^3A_{b=768/6}$) | 2E-1 | 3E-1 | 2E-1 | 7E-2 | 3E-1 | 2E-1 |
| | Learning Rate (Head) | 2E-4 | 4E-6 | 3E-2 | 8E-6 | 6E-3 | 4E-2 |
| Large | Epochs | 10 | 80 | 70 | 30 | 60 | 40 |
| | Batch Size | 128 | 128 | 128 | 32 | 64 | 128 |
| | Learning Rate ($C^3A_{b=1024/8}$) | 9E-2 | 3E-1 | 2E-1 | 7E-2 | 5E-2 | 2E-1 |
| | Learning Rate (Head) | 2E-4 | 5E-6 | 3E-3 | 8E-6 | 3E-3 | 5E-4 |

Table A4: Hyperparameter setup of LoRA and $C^3A$ for instruction tuning.

| Model | Hyperparameter | GSM8k | MATH | ViGGO | SQL | BoolQ | PIQA | SIQA |
|---|---|---|---|---|---|---|---|---|
| | Optimizer | | | | AdamW | | | |
| | LR Scheduler | | | | Cosine | | | |
| | Batch Size | | | | 16 | | | |
| | Warmup Ratio | | | | 0.05 | | | |
| | Dropout | | | | 0.05 | | | |
| | Epoch | | | | 3 | | | |
| LLaMA2-7B | Learning Rate (LoRA) | 5E-4 | 5E-4 | 5E-4 | 6E-4 | 5E-4 | 4E-4 | 6E-4 |
| | Learning Rate (DoRA) | 4E-4 | 5E-4 | 4E-4 | 6E-4 | 4E-4 | 5E-4 | 5E-4 |
| | Learning Rate ($C^3A$) | 8E-1 | 5E-1 | 5E-1 | 9E-1 | 7E-1 | 4E-1 | 3E-1 |
| LLaMA2-13B | Learning Rate (LoRA) | 5E-4 | 6E-4 | 5E-4 | 6E-4 | 5E-4 | 5E-4 | 4E-4 |
| | Learning Rate (DoRA) | 4E-4 | 6E-4 | 5E-4 | 6E-4 | 4E-4 | 4E-1 | 5E-1 |
| | Learning Rate ($C^3A$) | 6E-1 | 4E-1 | 8E-1 | 1 | 4E-1 | 4E-1 | 8E-1 |
| LLaMA2-8B | Learning Rate (LoRA) | 5E-4 | 5E-4 | 4E-4 | 5E-4 | 4E-4 | 4E-4 | 5E-4 |
| | Learning Rate (DoRA) | 6E-4 | 2E-4 | 5E-4 | 5E-4 | 4E-4 | 4E-4 | 4E-4 |
| | Learning Rate ($C^3A$) | 5E-1 | 3E-1 | 6E-1 | 4E-1 | 3E-1 | 3E-1 | 4E-1 |

Table A5: Hyperparameter setup of $C^3A$ for image classification tasks.

| Model | Hyperparameter | Pets | Cars | DTD | EuroSAT | FGVC | RESISC |
|---|---|---|---|---|---|---|---|
| Both | Optimizer | | | | AdamW | | |
| | LR Schedule | | | | None | | |
| | $C^3A$ Initialization | | | | Xavier Uniform | | |
| | Epochs | | | | 10 | | |
| | Batch Size | | | | 64 | | |
| Base | Learning Rate ($C^3A_{b=768/12}$) | 4E-1 | 4E+0 | 2E+0 | 2E+0 | 7E+0 | 2E+0 |
| | Learning Rate (Head) | 1E-2 | 1E-2 | 2E-2 | 8E-3 | 1E-2 | 2E-2 |
| | Weight Decay | 3E-4 | 5E-4 | 6E-5 | 2E-5 | 1E-5 | 2E-5 |
| Large | Learning Rate ($C^3A_{b=1024/16}$) | 7E-1 | 4E+0 | 2E+0 | 2E+0 | 4E+0 | 3E+0 |
| | Learning Rate (Head) | 3E-3 | 8E-3 | 7E-3 | 2E-2 | 1E-1 | 4E-3 |
| | Weight Decay | 4E-3 | 1E-5 | 2E-4 | 5E-4 | 2E-5 | 9E-5 |

## C  IMPLEMENTATIONS

---

**Algorithm A1** Block-Circular Convolution PyTorch Implementation

---

```python
import torch
from torch.autograd import Function
from torch.fft import fft, ifft

class BlockCircularConvolution(Function):
    @staticmethod
    def forward(ctx, x, w):
        m, n, b = w.shape
        x = x.reshape(*x.shape[:-1], n, b)
        ctx.save_for_backward(x, w)
        x = torch.einsum( "...nb,mnb->...mb", ifft(x), fft(w) )
        x = fft(x).real
        x = x.reshape(*x.shape[:-2], -1)
        return x

    @staticmethod
    def backward(ctx, grad_output):
        x, w = ctx.saved_tensors
        m, n, b = w.shape
        grad_output = grad_output.reshape(*grad_output.shape[:-1], m, b)
        grad_output_fft = fft(grad_output)
        x_grad = fft(torch.einsum( "...mb,mnb->...nb", grad_output_fft, ifft(w))).real
        x_grad = x_grad.reshape(*x_grad.shape[:-2], -1)
        w_grad = fft(torch.einsum( "...mb,...nb->mnb", grad_output_fft, ifft(x))).real
        return x_grad, w_grad
```

---

We present the PyTorch implementation of Block-Circular Convolution in Algorithm A1. Furthermore, due to the inefficiency of directly assigning entries (as shown in Equation 3), we derive an alternative algorithm to compute the $\Delta\mathbf{W}$ more efficiently. Rather than direct assignment, we employ a forward process on the Identity matrix. Mathematically, this can be expressed as

$$
\begin{aligned}
\Delta\mathbf{W} &= \mathcal{C}_{\text{blk}}(\Delta\mathbf{w}) \\
&= \mathcal{C}_{\text{blk}}(\Delta\mathbf{w}) \cdot \mathbf{I}_{d_2} \\
&= \mathcal{C}_{\text{blk}}(\Delta\mathbf{w}) \cdot [\mathbf{e}_1, \mathbf{e}_2, \cdots, \mathbf{e}_{d_2}] \\
&= [\mathcal{C}_{\text{blk}}(\Delta\mathbf{w})\mathbf{e}_1, \mathcal{C}_{\text{blk}}(\Delta\mathbf{w})\mathbf{e}_2, \cdots, \mathcal{C}_{\text{blk}}(\Delta\mathbf{w})\mathbf{e}_{d_2}] \\
&= [\Delta\mathbf{w} \star \mathbf{e}_1, \Delta\mathbf{w} \star \mathbf{e}_2, \cdots, \Delta\mathbf{w} \star \mathbf{e}_{d_2}].
\end{aligned}
$$

Where $\mathbf{I}_{d_2} \in \mathbb{R}^{d_2 \times d_2}$ represents an Identity matrix and $\mathbf{e}_i$ is the $i$th column of it. In pytorch, we can efficiently compute the iFFT of $\{\mathbf{e}_i\}_{i=1,2,\cdots,d_2}$ by a column-wise iFFT of $\mathbf{I}_{d_2}$. We present the Pytorch implementation in Algorithm A2 as well.

---

**Algorithm A2** Fast Algorithm of Getting $\Delta\mathbf{W}$

---

```python
import torch
from torch.fft import fft, ifft

def get_circulant_fast(w):
    m, n, b = w.shape
    x = torch.eye(n*b, dtype=w.dtype, device=w.device)
    x = x.reshape(*x.shape[:-1], n, b)
    x = torch.einsum( "...nb,mnb->...mb", ifft(x), fft(w) )
    x = fft(x).real.flatten(start_dim=1).T
    return x
```

---

# D  INSTRUCTION TUNING EXAMPLES

---

**LLaMA3-8B (MATH)**

Find the number of square units bounded by the $x$-axis, the $y$-axis and the line $x + y = 6$.
Answer the above question. First think step by step and then answer the final number.

**Response**

| Methods | Response |
|---|---|
| **LoRA** | The line $x + y = 6$ intersects the $x$-axis at $x = 6$ and the $y$-axis at $y = 6$. 
 So, the line divides the square with side length 6 into two triangles. 
 The area of each triangle is $\frac{1}{2} \times 6 \times 6 = 18$. 
 So, the area of the square is $2 \times 18 = \boxed{36}$. 
 The final answer is: 36 |
| **C³A** | The line $x + y = 6$ intersects the $x$-axis at $(6, 0)$ and the $y$-axis at $(0, 6)$. 
 Thus, the line divides the first quadrant into two triangles, one with base 6 and height 6 and the other with base 6 and height 0. 
 The area of the first triangle is $(6 \times 6)/2 = 18$ square units, and the area of the second triangle is $(6 \times 0)/2 = 0$ square units. 
 Thus, the total area is $18 + 0 = \boxed{18}$ square units. 
 The final answer is: 18 |

---

**LLaMA3-8B (GSM8K)**

Tim makes a special honey and jam mix. To make 1 jar of the mixture it takes 2 pounds of passion fruit and 1.5 pounds of jackfruit. It also takes 2 pounds of special honey. The passion fruit costs $6 per pound, the jackfruit is $8 per pound, and the honey is $10 per pound. He sells each jar for $50. How much profit does he make selling 10 jars?
Answer the above question. First think step by step and then answer the final number.

**Response**

| Methods | Response |
|---|---|
| **LoRA** | The passion fruit costs $6 \times 2 = 12$. 
 The jackfruit costs $8 \times 1.5 = 12$. 
 The honey costs $10 \times 2 = 20$. 
 So the total cost is $12 + 12 + 20 = 44$. 
 So he makes a profit of $50 \times 10 - 44 = 460$. 
 The final answer is: 460 |
| **C³A** | The passion fruit costs $6 \times 2 = 12$. 
 The jackfruit costs $8 \times 1.5 = 12$. 
 So the fruit costs $12 + 12 = 24$. 
 The honey costs $10 \times 2 = 20$. 
 So the total cost per jar is $24 + 20 = 44$. 
 So he makes a profit of $50 - 44 = 6$ per jar. 
 So he makes a profit of $6 \times 10 = 60$. 
 The final answer is: 60 |

