# OpenReview forum: "Parameter-Efficient Fine-Tuning   via Circular Convolution"
_ICLR.cc/2025/Conference — Submitted to ICLR 2025_

### Official Review · Reviewer_tV7B · 2024-10-16

**Soundness:** 2
**Presentation:** 4
**Contribution:** 1
**Rating:** 3
**Confidence:** 4

**Summary:**

This paper borrows the block circulant method from existing work and applies to LLMs. The proposed method is about using block circulant matrix for training which has been already published in the referenced work (Ding et al., 2017). However, the entire method writing (section 3) does not acknowledge/cite the work at all and instead present as if authors created block circulant based training. As a result, the major contribution narrows down to the empirical verification of block circulant training method on different large models.

**Strengths:**

1. Experimental results are very promising, and evaluated models cover both the base and large version of RoBERTa and ViT, also including  LLaMA-2 and LLaMA-3.
2. There is also synthetic data experiment which is simple and easy to demonstrate the effectiveness of proposed method against LoRA and full fine-tuning.

**Weaknesses:**

1. No technical novelty, because the proposed method is borrowed from the referenced work (Ding et al., 2017)
2. Lack of analysis regarding the impact of block size setting, e.g., smaller block size gets more parameters but smaller GPU cost (Table 2)? Why more parameters result in smaller GPU cost? According to theoretical space analysis (section 3.5.2), with d1=d2=768, the formulation cannot explain the observation.
3. FFT based solution is naturally sensitive to intialization, as a small change in frequence domain can be easily spreaded to all time domain. The experimental design in section 4.5 should include results on LLaMA models, as the FFT size is large in LLaMA models. The hypothesis is inclined to be that intialization for FFT based method is sensitive in the case of large models or large FFT size settings.

**Questions:**

1. Can you compare with the state-of-the-art FourierFT method (https://icml.cc/virtual/2024/poster/33821), which is also a Fourier domain based method?
2. Can you provide memory cost details in Table 3 and Table 4, as you did for Table 2?
3. Need a breakdown of the memory cost of proposed method, since in Table 2, more parameters somehow results in less GPU memory for both base and large models.
4. Initialization test for large FFT size setting like LLaMA for section 4.5
5. FFT operations can easily result in nan/inf values in low precision computation. Does the proposed method actually run in half precision (as LLMs are stored) or full precision?

**Details Of Ethics Concerns:**

The paper method comes from one of its cited papers, i.e., (Ding et al., 2017). To be more specific, that paper has already published the forward and backward propagation of block circulant matrix based training in deep learning community. Somehow, authors do not acknowledge/cite the work in their proposed method (section 3) while being aware of the paper. Instead, authors wrapped the method under the name of circular convolution and block circular convolution. As a result, the methodology writing (section 3) appears as if authors created the block circulant matrix based training for deep learning models.

---

> ### Author Response · Authors · 2024-11-19
> **Response to Reviewer tV7B - (1/2)**
>
> Thanks for your constructive comments! We try our best to address your concerns about the novelty and empirical evaluations.
>
> > Weakness 1: doubts on plagiarism and novelty.
>
> We appreciate our esteemed reviewer for pointing out this concern, and we have the following responses.
>
> * We would like to express our sincere acknowledgment of the pioneering work by Ding et al. (2017), who was the first to apply block circulant matrices in the deep learning community. We apologize for the unintentional omission of this important citation in our methodology section and will incorporate proper acknowledgment in the revised camera-ready version.
>
> * We wish to clarify that we do not claim to have developed new forward and backward algorithms for block circulant matrix-based learning. Our contribution lies in applying this existing approach within the context of PEFT. We appreciate the feedback regarding the perceived limitations of our paper's novelty and would like to offer further clarification. The fine-tuning scenario—where data scarcity necessitates regularization techniques—is fundamentally different from the standard training scenario. As discussed in lines 74–79, the evolution from CNNs to Transformers underscores the benefits of removing intrinsic biases (i.e., special patterns or algorithmic restrictions) when abundant data is available. This shift explains why model compression techniques, such as low-rank decomposition and circulant-matrix-based learning, have become less prominent in the era of large language models (LLMs), which are pretrained with a considerable amount of data. However, as we posit in lines 133–139, these methods remain invaluable within the PEFT community, where limited data is still a stumbling block and intrinsic biases are highly beneficial.
>
> * In fact, the PEFT community often draws inspiration from traditional deep learning research, especially from the model compression field. For example, the influential LoRA method focuses on low-rank decomposition, a topic that has been widely studied (e.g., [1,2]). Additionally, the Fourier Transformation-based method our reviewer mentioned has also been explored in [3]. While these traditional approaches have not fully thrived in the era of modern LLMs, their foundational contributions continue to offer valuable insights and benefits for advancing areas like PEFT.
>
> [1] Zhang, Xiangyu, et al. "Accelerating very deep convolutional networks for classification and detection." TPAMI, 2015.
>
> [2] Denton, Emily L., et al. "Exploiting linear structure within convolutional networks for efficient evaluation." In NeurIPS, 2014.
>
> [3] Koutnik, Jan, et al. "Evolving neural networks in compressed weight space." In GECCO, 2010.
>
> > Question 1: Comparison against FouirerFT.
>
> Thank you for your suggestion. We have conducted comparisons between C3A and FourierFT on both RoBERTa-Base and ViT-Base models. To ensure fair comparisons (i.e., a similar parameter size), we set $b = 768$ and $n = 1,000$ for RoBERTa-Base. For ViT-Base, we chose $b = 768/12 = 64$ and $n = 10,000$. The resulted accuracies are reported as follows:
>
> | Task | FourierFT | C3A |
> | :----|:----|:----|
> | CoLA | 61.44 | **61.83** |
> | STS-B | 89.45 | **90.46** |
> | Cars | 79.12 | 79.05 |
> | DTD | 79.27 | **80.57** |
>
> The parameter size is listed below:
>
> | Model | FourierFT | C3A |
> | :----|:----|:----|
> | RoBERTa-Base | 0.024M | **0.018M** |
> | ViT-Base | 0.24M | **0.22M** |
>
> From the result, we can observe that C3A generally outperforms FourierFT when constrained with similar parameter size.
>
> > Weakness 2 & Question 2, 3: Experiments and explanation on memory cost.
>
> We thank the reviewer for raising this valuable concern. We would like to correct our space complexity analysis as follows. On the GPU, the cuFFT backend automatically parallelizes the FFT operation along non-operated axes, with the parallelization size depending on the available resources. Therefore, the auxiliary tensor size should be $nb$ instead of $b$, where $n$ is the parallelization size. Consequently, the theoretical memory lower bound becomes $2\sqrt{n d_1 d_2}$ when $b^* = \sqrt{\frac{d_1 d_2}{n}}$. Given that $d_1 = d_2 = 768$, if $n > 1$, a larger $b$ could naturally result in lower memory consumption than when $b = d_1 = d_2 = 768$.
>
> As suggested, we have conducted an additional comparison of the actual memory cost between C3A and LoRA for Table 3 and Table 4. To ensure fairness, we follow the same experimental settings and set the sequence length to 256 for LLaMA families. The results are reported as follows:
>
> | Method | ViT-Base | ViT-Large | LLaMA2-7B | LLaMA2-13B | LLaMA3-8B |
> | :----|:----|:----| :----| :----| :----|
> | LoRA | 7.81 GB | 17.86 GB | 23.99 GB | 41.49 GB | 30.05 GB |
> | C3A | **7.24 GB** | **16.91 GB** | **23.05 GB** | **40.17 GB** | **29.02 GB** |
>
> We can observe that C3A continues to demonstrate lower memory consumption, consistent with the results observed in the GLUE benchmark.

---

> > ### Comment · Reviewer_tV7B · 2024-11-25
> >
> > Dear authors,
> >
> > Your feedback is important and sensitive, especially on the part of plagiarism and novelty. The promise on proper acknowledgement in revised version shall be appreciated. Unfortunately, current ratings and concerns, are and will be, based on paper upfront. Regarding a risky paper, my rating will not change, but it does not mean the overall decision on your paper.

---

> > > ### Author Response · Authors · 2024-11-28
> > >
> > > Dear esteemed reviewer tV7B,
> > >
> > > With the discussion period nearing its conclusion, we would like to take this opportunity to encourage further exchange of ideas regarding our paper.
> > >
> > > First and foremost, we want to emphasize that it was never our intention to overlook the contributions made by Ding et al. The concept of **block circulant matrices** has been an established part of linear algebra for years [1]. Given its longstanding presence as a mathematical tool, we believe that its usage does not necessarily require specific citation of application-oriented papers. However, if you believe a citation is essential in this context, [1] might be a more fitting reference.
> > >
> > > Furthermore, while it is true that **gradient calculation of block circulant matrices** has been explored in the work by Ding et al. and in [2], our contribution lies in simplifying the derivation process by leveraging the **commutative property of convolution**. This distinguishes our approach and underscores the originality of our work, which should not be misconstrued as plagiarism.
> > >
> > > That said, we do fully recognize the contributions of Ding et al. and have formally cited and acknowledged their work in the updated version of our manuscript. We hope this clarification addresses your concerns. We respectfully request that the review focus on other aspects of our work, **particularly on your other concerns we have been trying to address**, rather than reiterating the resolved "plagiarism" issue or regarding our paper as "risky".
> > >
> > > Thank you for your understanding and for facilitating constructive dialogue.
> > >
> > > [1] Rjasanow, Sergej. "Effective algorithms with circulant-block matrices." Linear Algebra and its applications 202 (1994): 55-69.
> > >
> > > [2] Cheng, Yu, et al. "An exploration of parameter redundancy in deep networks with circulant projections." Proceedings of the IEEE international conference on computer vision. 2015.

---

> > > > ### Comment · Area_Chair_1T9X · 2024-11-30
> > > >
> > > > Dear Reviewer tV7B,
> > > >
> > > > Thanks for your review particularly around the acknowledgement of Ding et al. 2017. With the existence and similar approach used in Ding et al. 17, do you think this paper's method still presents a notable contribution? Also, it would be great if you could comment on whether authors have addressed your other concerns.
> > > >
> > > > Thanks, AC

---

> ### Author Response · Authors · 2024-11-19
> **Response to Reviewer tV7B - (2/2)**
>
> > Weakness 3 & Question 4: Experiments on initialization.
>
> Thanks for your suggestion. We have incorporated an empirical comparison of different initialization strategies using LLaMA3-8B on GSM8K, SQL, and PIQA, which are the representative datasets of three instruction-tuning tasks. We have carefully tuned the hyperparameters to the best of our ability, and the results are presented below:
>
> | Task | Zero | Gaussian | Kaiming | Xavier | Range |
> | :----|:----|:----|:----|:----|:----|
> | GSM8K | 63.55 | 63.76 | 63.53 | 64.22 | 0.69 |
> | SQL | 80.58 | 80.38 | 80.45 | 80.73 | 0.35 |
> | PIQA | 90.59 | 90.26 | 90.12 | 90.33 | 0.47 |
>
> The observed variations among different initialization strategies are relatively insignificant, which is consistent with similar trends noted in the GLUE benchmark and image classification.
>
> > Question 5: Training details about precision.
>
> Thank you for bringing up this important issue. All our experiments were conducted using half-precision (BF16). Specifically, both the activations $\mathbf{x}$ and the pretrained weights $\mathbf{W}$ were initialized in BF16. However, due to the current lack of BF16 support in cuFFT, we temporarily converted $\mathbf{x}$ to FP32 for the circular convolution operation, then converted it back to BF16 afterward, maintaining the foundation model in BF16 throughout.

---

> ### Author Response · Authors · 2024-11-25
>
> Thank you for the feedback.
>
> We have updated the manuscript to address your concerns regarding the plagiarism issue. The revised parts are highlighted in magenta for your convenience. We would greatly appreciate any further feedback you may have.

---

### Official Review · Reviewer_VhRA · 2024-11-02

**Soundness:** 3
**Presentation:** 3
**Contribution:** 2
**Rating:** 6
**Confidence:** 4

**Summary:**

The paper proposes a new parameter-efficient fine-tuning (PEFT) method (denoted C³A) that leverages circulant matrices and Fast Fourier Transform (FFT) to achieve high-rank adaptations while maintaining computational efficiency. The proposed method allows independent control of the rank and number of trainable parameters through block-circular convolution. The authors evaluate their method across various NLP and vision tasks, including NLP, comparing the performance of C³A to existing PEFT.

**Strengths:**

1. Providing an interesting approach for low-rank adaptation that relies on FFT instead of the typical residual matrix to try to address existing limitations.
2. The extension to block-circular convolution allows for better control over parameter numbers (and makes the method applicable to non-square matrix weights) without compromising rank.
3. Providing detailed practical implementation and foundation for the different components (e.g., circular convolution)

**Weaknesses:**

1. While using circular convolution and FFT in PEFT is interesting, the improvement over LoRA and similar methods may be seen as incremental, with modest/marginal gains rather than a significant improvement over existing PEFT methodologies.
2. The method's reliance on FFT may introduce implementation complexity, particularly in environments with limited support for these operations, which could restrict the practical applicability of the method.
3. Many of the presented results don’t show clear gains or dominance, which questions the reliability of the presented method, which is the main weakness of the paper.

**Questions:**

1. Given that the results don’t dominate existing methods in many scenarios, what would be the ideal scenario/environment/conditions in which the proposed method is most effective?
2. Is there an ablation study for the block size (b)? I understand that it is constrained because of the divisibility, but I believe it is important to see some insights about its impact on the result.
3. Can you present more insights/analysis about the decision and the impact of using FFT (e.g., some ablation for the FFT part on its own)?

---

> ### Author Response · Authors · 2024-11-19
> **Response to Reviewer VhRA**
>
> Thank you for the provided comments and helpful suggestions! We have incorporated additional experiments and discussions, and we hope they adequately address your concerns.
>
> > Weakness 1, 3 & Question 1: Ideal scenarios for C3A.
>
> We thank the reviewer for highlighting this important issue. In practical applications of PEFT methods, there is an increasing demand for deployment in ever-changing scenarios and personalization. However, applying LoRA across multiple instantiations [1] presents challenges due to its large parameter size, particularly in terms of storage and transmission, especially for low-resource hardware and consumer-grade networks [2]. In contrast, we'd like to highlight that C3A achieves competitive performance while requiring not only fewer parameters but also reduced actual GPU memory. Therefore, we believe C3A offers a strong alternative for real-world deployment on resource-constrained devices.
>
> [1] Sheng, Ying, et al. "S-LoRA: Serving Thousands of Concurrent LoRA Adapters." In MLSys, 2024.
>
> [2] Borzunov, Alexander, et al. "Distributed Inference and Fine-tuning of Large Language Models Over The Internet." In NeurIPS, 2023.
>
> > Question 2: Ablation study on block size.
>
> Thank you for your valuable suggestion. We perform extensive evaluations of C3A with varying block sizes across several representative datasets. Specifically, tasks on CoLA and STS-B are evaluated using RoBERTa-Base, while Cars and DTD utilize ViT-Base. For larger-scale models, we conduct evaluations on GSM8K, SQL, and PIQA using LLaMA3-8B. The results are presented below:
>
> | Task | $b$=64 | $b$=128 | $b$=256 | $b$=768 |
> | :----|:----|:----|:----|:----|
> | CoLA | **62.48** | 62.07 | 61.15 | 61.83 |
> | STS-B | 90.16 | 89.98 | 90.29 | **90.46** |
> | Cars | **79.05** | 77.75 | 74.87 | 71.52 |
> | DTD | **80.57** | 78.04 | 77.89 | 77.32 |
>
> | Task | $b$=64 | $b$=128 | $b$=256 | $b$=512 |
> | :----|:----|:----|:----|:----|
> | GSM8K | **66.72** | 64.22  | 63.53 | 62.09 |
> | SQL | **80.81** | 80.73 | 80.07 | 79.54 |
> | PIQA | 90.05 | 90.33 | 90.21 | **90.42** |
>
> As observed, different tasks respond differently to changes in block size. Simple tasks like STS-B and PIQA perform well even with a very large block size, which corresponds to a lower parameter count. In contrast, more complex tasks such as Cars and GSM8K require a smaller block size to achieve optimal performance. Therefore, much like selecting the appropriate rank $r$ when using LoRA, we recommend searching for the optimal block size $b$ when approaching a new task.
>
> > Weakness 2 & Question 3: Practical support of FFT.
>
> Thank you for the instructive comments. We acknowledge the fact that FFT is a rather complex method to implement. However, we have observed that leading GPU manufacturers, including NVIDIA [1], AMD [2], and Intel [3], have provided easy-to-use implementations of FFT, which might alleviate the concern about the practicality.
>
> [1] https://docs.nvidia.com/cuda/cufft/index.html
>
> [2] https://rocm.docs.amd.com/projects/rocFFT/en/latest/
>
> [3] https://docs.openvino.ai/2022.3/openvino_docs_MO_DG_prepare_model_Supported_Frameworks_Layers.html

---

> > ### Author Response · Authors · 2024-11-26
> >
> > Dear Reviewer VhRA,
> >
> > Thank you for your time and the valuable insights you’ve provided on our work. As we near the conclusion of the author-reviewer discussion period, we truly appreciate the opportunity to address your concerns.
> >
> > We kindly request you to review our responses to your comments to ensure they adequately address the points you raised. If you have any additional questions or need further clarification, please don’t hesitate to reach out—we’d be happy to provide more details.
> >
> > Best,
> >
> > Authors

---

> > > ### Comment · Reviewer_VhRA · 2024-11-26
> > >
> > > Thank you for your responses, while I am still skeptical about the strength of the gains from the proposed method, the rest of my concerns have been addressed and I will take it into consideration for the final score.

---

> > > > ### Author Response · Authors · 2024-11-27
> > > >
> > > > We heartfully thank you for your time and effort in reviewing our paper and response! And we always remain eager to address any further questions or concerns you may have.

---

> ### Author Response · Authors · 2024-11-25
>
> Dear reviewer VhRA,
>
> Thank you once again for taking the time and effort to provide valuable feedback on our paper. We have carefully addressed your concerns by adding discussions and additional experiments. We would greatly appreciate the opportunity to discuss whether there are any remaining issues or concerns you would like us to address.
>
> Thank you, Authors

---

### Official Review · Reviewer_F24A · 2024-11-04

**Soundness:** 3
**Presentation:** 3
**Contribution:** 3
**Rating:** 6
**Confidence:** 4

**Summary:**

This work introduces Circular Convolution Adaptation (C3A) as an advanced parameter-efficient fine-tuning (PEFT) technique, addressing limitations of existing methods like Low-Rank Adaptation. C3A leverages the circular convolution operator, enabling high-rank adaptation without a proportional increase in trainable parameters. Utilizing Fast Fourier Transform (FFT) operations, C3A achieves efficient memory and computational performance while retaining expressive power. The method effectively reduces memory requirements and computational load compared to high-parameter alternatives like VeRA. A notable strength of C3A is its flexibility in adjusting the number of trainable parameters through block-circular convolution, which adapts to diverse downstream tasks. Experimental results validate C3A's superior accuracy and memory efficiency across various tasks.

**Strengths:**

1. High-Rank Adaptation with Low Parameters: C3A achieves high-rank adaptation without increasing the number of trainable parameters. By leveraging circular convolution, it avoids the limitations of low-rank adaptation in methods like LoRA, maintaining expressive power with fewer resources.
﻿
2. Efficient Use of FFT for Computational Gains. The use of Fast Fourier Transform (FFT) in both forward and backward propagation enhances computational speed and reduces complexity, achieving a time complexity comparable to LoRA but with better efficiency in memory use.
﻿
3. Memory Savings with Block-Circular Convolution. By implementing block-circular convolution, C3A allows for flexible control of trainable parameters. This adaptability to block size enables parameter tuning based on task requirements, optimizing both memory and computational resources.
﻿
4. Comprehensive Benchmark Performance. C3A demonstrates robustness performance across multiple tasks, including GLUE benchmarks for NLP and Vision Transformer (ViT) for image classification, showing significant improvements in accuracy with fewer parameters.
﻿
5. Adaptable Parameter Count with Minimal Performance Trade-Offs. C3A allows for adjustable parameter configurations through its block-circular design, offering a range of parameter counts without significant sacrifices in performance, providing versatility across tasks.

**Weaknesses:**

1. Although C3A claims robustness to initialization methods, the paper does not thoroughly analyze how different initializations may impact convergence speed or model performance across varying tasks. Additional experiments on initialization sensitivity could further validate this robustness claim and clarify any subtle effects on fine-tuning stability.
﻿
2. While the method demonstrates efficiency on moderate NLP and CV benchmarks, it lacks evaluation on tasks with substantially higher dimensions or complexity, where circular convolution’s rank flexibility may not be as effective. Testing C3A on such high-dimensional data would strengthen claims about its scalability.
﻿
3. Inadequate Justification for Block Size Selection: The paper introduces block-circular convolution with adaptable block sizes but provides minimal guidance on how to select optimal block sizes for specific tasks. This omission makes it challenging for readers to understand the trade-offs and practical considerations in choosing block sizes to balance accuracy and parameter efficiency.
﻿
6. The proposed method relies on block-circular convolution for flexibility, but the paper does not clearly address its limitations, such as potential rank deficiencies or performance degradation in non-square matrices. Expanding on these limitations and detailing their implications would provide a more balanced view of C3A’s adaptability across different architectures.

**Questions:**

1. I will appreciate additional analysis on how initialization affects model convergence speed and accuracy across varied tasks. Specifically, experiments with detailed metric comparisons for convergence time under different initializations would strengthen this claim.

2. The method lacks experimental validation on tasks involving significantly higher dimensions or complexity, where circular convolution may face generalization challenges. We encourage the authors to include experiments on larger-scale datasets or more complex tasks to confirm C3A's adaptability and performance.

3. How different block sizes may impact both parameter efficiency and performance across tasks? We request the authors to provide guidelines for block size selection.

4. Aside from the existing comparison with baseline methods LoRA and VeRA, we suggest authors to Include more recent PEFT methods as baselines, as that will provide a more comprehensive understanding of C3A's standing relative to broader industry benchmarks.

5. Please elaborates the limitations of the proposed Block-Circular Convolution.

---

> ### Author Response · Authors · 2024-11-19
> **Response to Reviewer F24A - (1/2)**
>
> We really appreciate the reviewer for listing the detailed strengths, which have clearly recognized the novelty and contributions of our paper. As requested by the reviewer, extensive additional experiments are conducted to further support our contributions.
>
> > Question 1: Convergence analysis of different initializations.
>
> Thanks for raising this question. As suggested by the reviewer, we have collected data on the convergence time, which is represented by the percentage of training progress, for 4 initialization methods. The results are presented as follows:
>
> | Task | Zero | Gaussian | Kaiming | Xavier |
> | :----|:----|:----|:----|:----|
> | CoLA | 12% | 11% | 13% | 10% |
> | STS-B | 15% | 18% | 16% | 16% |
> | Cars | 90% | 91% | 89% | 91% |
> | DTD | 30% | 30% | 32% | 30% |
>
> The results observed indicate that different initialization strategies display similar convergence speeds, further supporting C3A's resilience to variations in initialization.
>
> > Question 2: Additional evaluations on larger-scale datasets or more complex tasks. (More LLM Datasets.)
>
> Thanks for the suggestions. For comprehensiveness, we further assess the adaptability of C3A on a more complex and larger scale Commonsense170K dataset [1], a widely used commonsense reasoning benchmark that is framed as multiple-choice questions. Specifically, it integrates the training sets of eight distinct datasets, including BoolQ, PIQA, SIQA, HellaSwag, WinoGrande, ARC-e, ARC-c and OBQA. The evaluations are then conducted on the test sets of the individual datasets.
>
> We have conducted a comparison of C3A against LoRA and DoRA using the LLaMA3-8B model. For consistency, we set $b = 128$ for C3A and $r = 32$ for the rest, ensuring an identical parameter count as outlined in Table 3. We try our best to tune the models and the results are presented as follows:
>
> | Methods | BoolQ | PIQA | SIQA | HellaS. | WinoG. | ARC-e | ARC-c | OBQA | Avg. |
> | :---- | :---- | :---- | :---- | :---- | :---- | :---- | :---- | :---- | :---- |
> | LoRA | 70.8 | 85.2 | 79.9 | 91.7 | 84.3 | 84.2 | 71.2 | 79.0 | 80.8 |
> | DoRA | 74.6 | 89.3 | 79.9 | 95.5 | 85.6 | 90.5 | 80.4 | 85.8 | 85.2 |
> | C3A | 74.5 | 88.7 | 79.8 | 95.2 | 85.2 | 90.7 | 79.7 | 86.0 | 85.0 |
>
> As observed, C3A continues to demonstrate competitive performance while consuming only half the parameter size, highlighting the effectiveness of circular convolution in managing generalized and complex tasks.
>
> [1] Hu, Zhiqiang, et al. "LLM-Adapters: An Adapter Family for Parameter-Efficient Fine-Tuning of Large Language Models." In EMNLP, 2023.
>
> > Question 3: Ablation study on block size.
>
> Thank you for your valuable suggestion. We performed extensive evaluations of C3A with varying block sizes across several representative datasets. Specifically, tasks on CoLA and STS-B were evaluated using RoBERTa-Base, while Cars and DTD utilized ViT-Base. For larger-scale models, we conducted evaluations on GSM8K, SQL, and PIQA using LLaMA3-8B. The results are presented below:
>
> | Task | $b$=64 | $b$=128 | $b$=256 | $b$=768 |
> | :----|:----|:----|:----|:----|
> | CoLA | **62.48** | 62.07 | 61.15 | 61.83 |
> | STS-B | 90.16 | 89.98 | 90.29 | **90.46** |
> | Cars | **79.05** | 77.75 | 74.87 | 71.52 |
> | DTD | **80.57** | 78.04 | 77.89 | 77.32 |
>
> | Task | $b$=64 | $b$=128 | $b$=256 | $b$=512 |
> | :----|:----|:----|:----|:----|
> | GSM8K | **66.72** | 64.22  | 63.53 | 62.09 |
> | SQL | **80.81** | 80.73 | 80.07 | 79.54 |
> | PIQA | 90.05 | 90.33 | 90.21 | **90.42** |
>
> As observed, different tasks respond differently to changes in block size. Simple tasks like STS-B and PIQA perform well even with a very large block size, which corresponds to a lower parameter count. In contrast, more complex tasks such as Cars and GSM8K require a smaller block size to achieve optimal performance. Therefore, much like selecting the appropriate rank $r$ when using LoRA, we recommend searching for the optimal block size $b$ when approaching a new task.
>
> > Question 4: Additional baselines.
>
> Thank you for your suggestion. We have conducted comparisons between C3A and FourierFT [1] on both RoBERTa-Base and ViT-Base models. To ensure fair comparisons (i.e., a similar parameter size), we set $b = 768$ and $n = 1,000$ for RoBERTa-Base. For ViT-Base, we chose $b = 768/12 = 64$ and $n = 10,000$. The resulted accuracies are reported as follows:
>
> | Task | FourierFT | C3A |
> | :----|:----|:----|
> | CoLA | 61.44 | **61.83** |
> | STS-B | 89.45 | **90.46** |
> | Cars | 79.12 | 79.05 |
> | DTD | 79.27 | **80.57** |
>
> The parameter size is listed below:
>
> | Model | FourierFT | C3A |
> | :----|:----|:----|
> | RoBERTa-Base | 0.024M | **0.018M** |
> | ViT-Base | 0.24M | **0.22M** |
>
> From the result, we observe that C3A generally outperforms FourierFT when constrained with similar parameter size.
>
> [1] Gao, Ziqi, et al. "Parameter-Efficient Fine-Tuning with Discrete Fourier Transform." In ICML, 2024.

---

> ### Author Response · Authors · 2024-11-19
> **Response to Reviewer F24A - (2/2)**
>
> > Question 5: Discussion on limitations.
>
> We thank the reviewer for this valuable question. We acknowledge that a limitation of C3A is the absence of an easy-to-use serving framework, such as S-LoRA [1]. However, given its effectiveness and efficiency, we believe that developing a serving framework for C3A is a promising and worthwhile direction for future exploration.
>
> [1] Sheng, Ying, et al. "S-LoRA: Serving Thousands of Concurrent LoRA Adapters." In MLsys, 2024.

---

> ### Author Response · Authors · 2024-11-25
>
> Dear reviewer F24A,
>
> Thank you once again for taking the time and effort to provide valuable feedback on our paper. We have carefully addressed your concerns by adding additional experiments and revisions. We would greatly appreciate the opportunity to discuss whether there are any remaining issues or concerns you would like us to address.
>
> Thank you, Authors

---

> ### Author Response · Authors · 2024-11-26
>
> Dear Reviewer F24A,
>
> Thank you for your time and the valuable insights you’ve provided on our work. As we near the conclusion of the author-reviewer discussion period, we truly appreciate the opportunity to address your concerns.
>
> We kindly request you to review our responses to your comments to ensure they adequately address the points you raised. If you have any additional questions or need further clarification, please don’t hesitate to reach out—we’d be happy to provide more details.
>
> Best,
>
> Authors

---

> ### Author Response · Authors · 2024-11-28
>
> Dear reviewer F24A,
>
> Thank you for your valuable contributions to the review process. As the deadline for the author-reviewer discussion period approaches, we sincerely appreciate the opportunity to address your concerns.
>
> We understand that you have a multitude of responsibilities. To facilitate a swift evaluation of our responses, we have summarized the corresponding changes as follows:
>
> * **Convergence Analysis:** We added the analysis of convergence across different initialization strategies ([Q1](https://openreview.net/forum?id=IZjBfdVRB0&noteId=kHWUEMWLEl)).
> * **Empirical Evaluations:** We expanded the evaluations to include larger-scale and more complex tasks ([Q2](https://openreview.net/forum?id=IZjBfdVRB0&noteId=kHWUEMWLEl)) and comparison against more recent baseline ([Q4](https://openreview.net/forum?id=IZjBfdVRB0&noteId=kHWUEMWLEl)).
> * **Ablation Study:** We added an ablation study on the block size ([Q3](https://openreview.net/forum?id=IZjBfdVRB0&noteId=kHWUEMWLEl)).
>
> We kindly request your feedback on whether our responses adequately address your concerns. If you have any further questions or additional points for discussion, please let us know—we would be happy to provide further clarification or additional details.
>
> Thank you for your time and consideration!
>
> Best,
>
> Authors

---

> ### Comment · Area_Chair_1T9X · 2024-11-30
>
> Dear Reviewer,
>
> Could you kindly respond and indicate whether authors have addressed your concerns?
>
> Thanks, AC

---

### Author Response · Authors · 2024-11-23
**General Response to All Reviewers**

Dear Reviewers,

We sincerely thank all the reviewers (`tV7B`, `VhRA`, `F24A`) for their valuable feedback. We glad that the reviewers apprecitated efficient incorporation of FFT in PEFT (`VhRA`, `F24A`), flexibility of block-circulant convolution (`VhRA`, `F24A`) and the extentiveness and solidness of our experimental evaluations (`F24A`, `tV7B`).

We have made every effort to faithfully address your comments in the responses. As suggested by the reviewers, we add

* Detailed justifications clarifying doubts related to plagiarism and novelty ([tV7B](https://openreview.net/forum?id=IZjBfdVRB0&noteId=dqnHXxnUgN)).
* Detailed explanations and additional experiments on memory cost ([tV7B](https://openreview.net/forum?id=IZjBfdVRB0&noteId=dqnHXxnUgN)).
* Additional empirical evaluations on larger-scale and more complex task ([F24A](https://openreview.net/forum?id=IZjBfdVRB0&noteId=HylF0fz28B)).
* Additional ablation study on block size ([F24A](https://openreview.net/forum?id=IZjBfdVRB0&noteId=HylF0fz28B), [VhRA](https://openreview.net/forum?id=IZjBfdVRB0&noteId=aEqI46VrD6)) and initialization strategies on large-scale LLaMA model ([F24A](https://openreview.net/forum?id=IZjBfdVRB0&noteId=kHWUEMWLEl)).
* Additional comparison againt more recent baseline FouierFT ([F24A](https://openreview.net/forum?id=IZjBfdVRB0&noteId=HylF0fz28B), [tV7B](https://openreview.net/forum?id=IZjBfdVRB0&noteId=dqnHXxnUgN)).
* Additional analysis of convergence across different initialization strategies ([F24A](https://openreview.net/forum?id=IZjBfdVRB0&noteId=kHWUEMWLEl)).
* Discussion on the ideal scenarios of C3A ([VhRA](https://openreview.net/forum?id=IZjBfdVRB0&noteId=aEqI46VrD6)).

We have made every effort to address all the concerns raised by our reviewers and remain committed to addressing any additional feedback that may arise. Furthermore, we are actively implementing the suggested revisions to prepare the updated version.

Thanks for all reviewers' time again.

Best regards,

Authors

---

### Author Response · Authors · 2024-11-26
**New Revision Reminder**

Dear Esteemed Reviewers,

We would like to kindly remind you of the approaching deadline for PDF revision and inform you that we have uploaded a new revision of our manuscript. This revision primarily includes the following updates:

1. **Acknowledgment and Citation**: The work by Ding et al. has been appropriately acknowledged and cited in the Methods section.
2. **Correction of Complexity Analysis**: Revisions have been made to clarify and correct the complexity analysis in the Methods section.

We sincerely appreciate your valuable time and effort in reviewing our work and look forward to your feedback.

Thank you once again for your support and consideration.

Best regards,
Authors

---

### Comment · Area_Chair_1T9X · 2024-11-28
**Reviewers, please kindly respond**

Dear Reviewers,

If you have not responded to author's rebuttal, please kindly do so as soon as possible. The deadline is Dec 2, but the authors can potentially further clarify questions if you respond earlier. Thanks!

Best, AC

---

### Meta-Review · Area_Chair_1T9X · 2024-12-08

**Metareview:**

(a) Summary

The paper proposes to use circular convolution as an alternative to the low-rank AB matrices in LoRA/PEFT. The method is called Circular Convolution Adaptation (C3A). It maintains higher rank than LoRA, with more efficiency benefits. It also outperforms LoRA variants.

(b) Strengths

Consistent performance improvement; higher-rank adaptation than LoRA; improved efficiency

(c) Weaknesses

Limited experiments with large models; more complexity (including initialization methods) than other LoRA variants (hence potential limited usage); not applicable to non-square matrix; marginal improvement in most cases

(d) Reasons for decision

Not enough support from reviewers; higher complexity outweighs the marginal benefits; lack of experiments on large models.

**Additional Comments On Reviewer Discussion:**

Improvement consistency
- new experiments provided, including one 8B model; concerns remain with limited large models and settings.

Initialization sensitivity
- new experiments provided, concerns mostly addressed

Applicability
- examples of NLP and CV tasks are provided, but broader validation is needed.

---

### Decision · Program_Chairs · 2025-01-22

Reject